**Subject Category:**
Biology (whole organism)

ecology/biomechanics

locomotion, diving, intermittent, buoyancy, biologging

**Author for correspondence:**
Adrian C. Gleiss
e-mail: a.gleiss@murdoch.edu.au

# Direct measurement of swimming and diving kinematics of giant Atlantic bluefin tuna (*Thunnus thynnus*)

Adrian C. Gleiss[1,2,3], Robert J. Schallert[1], Jonathan J. Dale[1], Steve G. Wilson[1] and Barbara A. Block[1]

[1]Tuna Research and Conservation Centre, Hopkins Marine Station, Stanford University, 120 Oceanview Boulevard, 93950 Pacific Grove, USA
[2]Centre for Sustainable Aquatic Ecosystems, Harry Butler Institute, and [3]College of Science, Health, Engineering and Education, Environment and Conservation Sciences, Murdoch University, 90 South Street, Murdoch, Western Australia 6150, Australia

ACG, 0000-0002-9960-2858

Tunas possess a range of physiological and mechanical adaptations geared towards high-performance swimming that are of considerable interest to physiologists, ecologists and engineers. Advances in biologging have provided significant improvements in understanding tuna migrations and vertical movement patterns, yet our understanding of the locomotion and swimming mechanics of these fish under natural conditions is limited. We equipped Atlantic bluefin tuna (*Thunnus thynnus*) with motion-sensitive tags and video cameras to quantify the gaits and kinematics used by wild fish. Our data reveal significant variety in the locomotory kinematics of Atlantic bluefin tuna, ranging from continuous locomotion to two types of intermittent locomotion. The tuna sustained swimming speeds in excess of $1.5 \text{ m s}^{-1}$ (0.6 body lengths $\text{s}^{-1}$), while beating their tail at a frequency of approximately 1 Hz. While diving, some descents were entirely composed of passive glides, with slower descent rates featuring more gliding, while ascents were primarily composed of active swimming. The observed swimming behaviour of Atlantic bluefin tuna is consistent with theoretical models predicting such intermittent locomotion to result in mechanical and physiological advantages. Our results confirm that Atlantic bluefin tuna possess behavioural specializations to increase their locomotory performance, which together with their unique physiology improve their capacity to use pelagic and mesopelagic habitats.

# 1. Introduction

Tunas are a clade of endothermic fish known for their elevated metabolic rates and unique swimming performance [1–3]. In addition to their endothermic tissues, tunas also feature morphological adaptations for high-performance swimming, such as a lunate tail and a musculoskeletal system transferring power from the anteriorly located slow-twitch 'red' muscle fibres to the tail. This mechanical arrangement results in a stiff swimming style, called thunniform locomotion [1,2,4,5]. Bluefin tunas, with their temperate and subpolar distribution of adult fish, feature a unique cardiac physiology and high aerobic capacity, given the cold temperatures they inhabit [6–8]. The physiological and biomechanical specializations are associated with extraordinary migratory behaviour of bluefin tunas, which regularly undertake transoceanic and rapid latitudinal migrations from often temperate productive foraging grounds to warmer spawning areas [9–11]. Due to their large size as adults, the large distances travelled in pelagic habitats have provided a significant challenge to studying the locomotion of bluefin tuna. Yet, understanding the biomechanics and locomotor activity of bluefin tunas is crucial, given their high metabolic rates and patchily distributed prey in the largely oligotrophic open ocean [12,13].

Despite the great interests in the biomechanical and physiological adaptations of tuna swimming [14], their kinematics have historically been studied in swim-tunnels and small tanks, putting severe constraints on the way these oceanic fish are able to move. This has resulted in the study of juvenile fish and may fail to highlight the complexity in their locomotor behaviour [15], especially since fish swimming under forced conditions can often display significantly different locomotor patterns compared to those swimming volitionally [16]. Theoretical models have long shown that different forms of intermittent locomotion, where animals intersperse active swimming with passive gliding can result in energetic advantages, reducing the mechanical work of swimming by up to 50% [17,18]. Considering that locomotion may comprise approximately 50% of routine metabolism in tuna [2,19], it is imperative to understand locomotor activity of wild fish to refine estimates of energy demand obtained from swim-tunnels [20].

The advent of motion-sensitive tags (accelerometers, magnetometers, gyroscopes) and their application in studying the movement of free-ranging animals [21–23] has removed these barriers. Recent work has revealed movement strategies in accordance with these predictions in a few select taxa; whale sharks (*Rhincodon typus*) and white sharks (*Carcharodon carcharias*) use passive glides to descend, while actively swimming with thrusts of the tail during ascents [24] to optimize their locomotion due to negative buoyancy, while two species of deep-sea shark do the opposite due to positive buoyancy [25]. Both scenarios corroborate Weihs' model of two-stage locomotion [17], by demonstrating that many marine animals use buoyant forces to glide and reduce mechanical work performed during vertical movements, with some notable exceptions [26,27]. Whale sharks and oceanic whitetip sharks have been shown to use diving geometries that minimize the cost of transport [28,29] and to shift diving behaviour in response to environmental conditions [30].

Despite the great promise of new biologging technologies, its uptake in the study of teleost fishes has been lagging behind the work on air-breathing marine vertebrates and elasmobranch fishes [24,25,31,32]. Indeed, due to their large size and prominent dorsal fins, much of the work on motion-sensitive biologging tags has come from elasmobranch fishes due to the relative ease of attaching data-loggers [33–37], whereas the morphology of pelagic teleost fish provides extraordinary challenges for firm attachment of loggers. Early work with multi-sensor acoustic tags has demonstrated some success in large teleost fishes, such as blue marlin (*Makaira nigricans*). Blue marlin with speed sensors were found to use two gears, swimming at low speeds of $0.15–0.25 \, \mathrm{m \, s^{-1}}$, with distinct periods of higher activity and speeds of $0.8–1.2 \, \mathrm{m \, s^{-1}}$ [38], considerably slower than recent estimates of swimming speeds by sailfish (*Istiophorus platypterus*), that cruise at approximately $2 \, \mathrm{m \, s^{-1}}$ and swim in bursts in excess of $6 \, \mathrm{m \, s^{-1}}$ [39]. In addition, new information on diving rates has come from archival tags placed on free-ranging tunas. These tags have relatively low temporal resolution (1–60 s) for capturing kinematic events and they are primarily programmed for long-term deployments to acquire sensor data to estimate geographical position and vertical behaviours for extended periods (months–years) until the fish is captured and the tag is recovered. The archival datasets have shown to date that tunas have a variable diving behaviour and respond to changing environmental conditions [40–43]. Remarkable dives have been measured with vertical velocities exceeding $6–8 \, \mathrm{m \, s^{-1}}$ and to depths exceeding 2000 m.

Archival tagging data have highlighted two prominent forms of vertical movement present in most species of pelagic fish; diel vertical migrations (DVMs) and repeated oscillatory diving [24,44–46]. Whereas the diel changes in vertical distribution of many species of tuna (and other pelagic predators)

were shown to relate to the diel changes in the distribution of prey, such arguments fail to explain repeated diving from the surface to depth. The use of archival or acoustic tags measuring temperature both in the viscera and ambient water temperatures has convincingly demonstrated thermoregulation as a driver of repeated vertical excursions, with particularly convincing examples for big-eye tuna (*Thunnus obesus*). Here, tuna perform rapid ascents up through the thermocline to rewarm in surface waters before returning to forage below the thermocline [47,48]. Similar patterns were recently demonstrated for *Mola mola* [49]. Whereas significant evidence has been amassed to support thermoregulation and prey searching as important factors shaping the repeated diving behaviour of tunas, little evidence (but see [50]) supports the hypotheses that dives are performed to improve locomotor performance, and by extension reduce energy expenditure, despite many researchers suggesting it may be an important factor [50].

The increasing use of accelerometers on fish has revealed how different ecological circumstances impact the swimming kinematics of fishes. Deployment of short-term accelerometry tags deployed on dolphin fish (*Coryphaena hippurus*) and anguillid eels (*Anguilla celebesensis* and *Anguilla marmorata*) revealed similar patterns to sharks in their diving behaviour [45,51]. Dolphin fish were found to descend by gliding, with the proportion of passive descents related to mixed layer depth; this was suggested to be an adaptation for efficient foraging [45]. Equally, anguillid eels performed repetitive diving with passive shallow descents and active steeper ascents. The function of these dives, however, was suggested to primarily relate to prepare fish for their long migration without wasting substantial amounts of energy [51]. However, much remains to be learned about teleost fishes and high-performance tunas in particular (but see [45,52,53]). Some of these gaps have recently been filled through experiments in captivity. Computational and experimental fluid dynamics have demonstrated the potential of passive gliding to decrease the horizontal cost of transport of captive Pacific bluefin tuna (*Thunnus orientalis*) [54]. In a separate study using biologging technology, Noda *et al.* [55] demonstrated smaller fish in mixed size Pacific bluefin tuna schools benefited less from intermittent swimming compared to larger fish within the same school.

Here, we report the first direct measurements on the locomotor activity of free-ranging Atlantic bluefin tuna (*Thunnus thynnus*) in the Gulf of St Lawrence equipped with animal attached accelerometers and cameras using a novel attachment technique for biologging tags. Our data confirm that Atlantic bluefin tuna possess considerable flexibility in their locomotor patterns, consistent with theoretical models predicting biomechanical advantages over continuous locomotion.

# 2. Methods

All work was performed in accordance with Stanford University's Institutional Animal Care and Use Committee (Protocol 13865) and satellite tagging was performed under permit by the Canadian Department of Fisheries and Oceans (permit no. SG-GNS-14-132A).

## 2.1. Deployment of tags

The electronic tagging experiments were conducted in the waters of the Gulf of St Lawrence out of Port Hood on Cape Breton Island, Nova Scotia, in October 2014 (electronic supplementary material, figure S1). The Atlantic bluefin tuna were caught on rod and reel (Penn 130) with fresh mackerel or herring baits. The terminal tackle was 300 lb leader with a circle hook. Tuna caught on hook and line were reeled in to the back of the vessel and brought onboard the tagging vessel using a specially designed titanium lip-hook that enabled pulling the fish through the boat's transom door onto a vinyl mat that was slick and wet. The use of a wet vinyl mat permitted the fish to slide easily without any friction or damage to the body. A saltwater hose was inserted immediately into the fish's mouth to oxygenate the gills. A soft cloth soaked in a fish protectant solution (PolyAqua®) was placed over the eyes to keep the fish calm. Fish were measured for curved fork length (CFL), sampled for genetics, tagged and released within 1–2 min of capture. Pictures of tag position were obtained upon release (figure 1).

## 2.2. Tag specifications and deployment

Once the tuna was on deck and well irrigated, tags were attached by placing two small holes (1 cm diameter) into the bony portion of the fin using a template corresponding to grooves in the tag and a small drill. The deployment techniques differed slightly between the two tag types used (customized

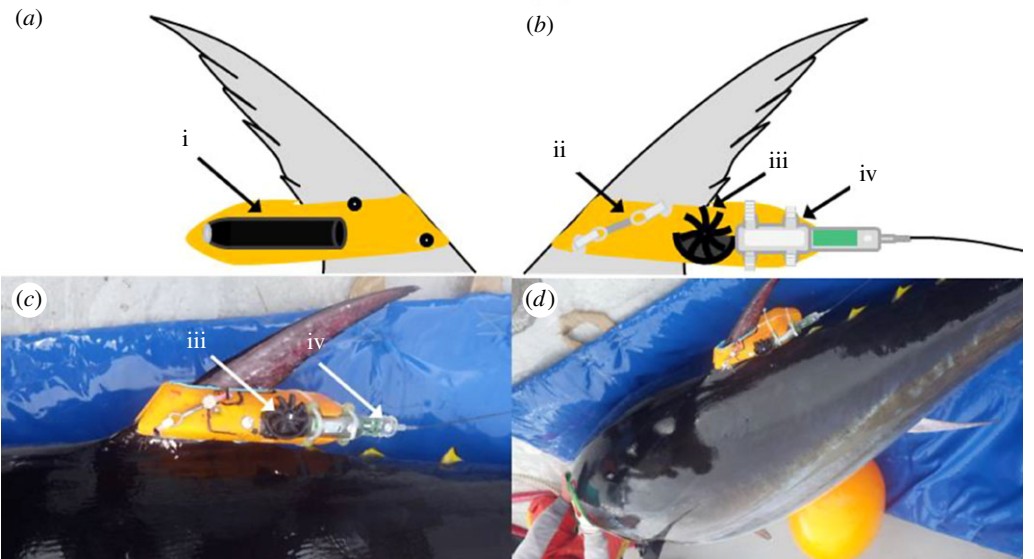

**Figure 1.** Deployment of a CATS video-data-logging tag to an Atlantic bluefin tuna. (*a*) Schematic of the tag attached to the second dorsal fin of a bluefin tuna showing the camera (i) attached to the right side of the orange floatation body containing the data-logging system. (*b*) The left side of the same tag showing the galvanic timed release (GTR) (ii) incorporated into the housing. (*c*) The same side of the tag attached to a live tuna prior to deployment showing fully assembled tag including paddle-wheel for speed measurement (iii) and smart position only tag (SPOT, iv) used for the relocation of the tag after corrosion of the GTR. (*d*) Tag placed on tuna ready for release. Tag attachment for fish no. 2 is not shown, since the tag design and application closely follows the one described by Whitmore *et al.* [33] for sharks and was simply transferred to tuna.

animal tagging solutions (CATS) tag (see figure 1) and a CEFAS G6a (see [33] and table 1 for tag sensor details). Both types of tags were encased in positively buoyant foam to permit floatation following pop-off, potentially altering buoyancy of its carrier. During the design process, excessive positive buoyancy was minimized to reduce this effect. Completely assembled tags only showed marginal positive buoyancy as evidenced by only a very small fraction of the tags protruding above the water's surface while floating.

## 2.2.1. CATS tag

The electronics were contained within a three-dimensional printed housing filled with a buoyant and pressure-resistant microspheres–epoxy mixture. Swimming speed was measured through an external paddle-wheel also constructed out of three-dimensional printed plastic. The rear-tip of the tag contained a slot that was designed to fit a Wildlife Computers SPOT tag (Wildlife Computers, Redmond, WA, USA). We adjusted the tag buoyancy such that the tag floated above the water surface after the tag popped off. The CATS tags of this first deployment were specifically designed for giant Atlantic bluefin tuna to mount on the second dorsal fin, maintaining the bulk package behind the fin, reducing their profile drag. The anterior part of the tag contained an open slot with a width of the chord thickness of the base of the second dorsal fin of an Atlantic bluefin tuna, the inside of the slot was also lined with a hypoallergenic foam (5 mm Poron Blue, Algeos, UK). Measurements were obtained from the second dorsal fin of fish caught by commercial fishermen prior to designing the mount for all tags. Two small holes were drilled into the fin (diameter less than 10 mm) using a template, and the tag was slid onto the fin. Two zip-ties attached to the two eye-holes in a galvanic timed release (GTR, http://www.neptunemarineproducts.com/galvanic-timed-releases/) were fed through the holes and sat in two 1 cm deep grooves of the tag. Both zip-tie ends were then fed through the two holes drilled in the fin using a template and fixed using the open case of a third zip-tie glued to a 4 cm outer diameter, 1.5 cm inner diameter nylon washer (Home Depot, USA). The surface of the buoyancy housing and the sides of the nylon washers in contact with the fin were also lined with hypoallergenic foam (3 mm thickness, PoronBlue, Algeos, UK) to avoid chafing. After corrosion of the GTR, the zip-ties were expected to loosen and the tag release and float to the surface. CATS tags sampled tri-axial acceleration, depth and temperature (and additional other sensor data not used here) at 40 Hz. In addition to its suit of sensors, the CATS tags contained a digital camera facing

**Table 1.** Details of all four Atlantic bluefin tuna tagged as part of this study.

| | tag date | time tagged | latitude | longitude | CFL (cm)[a] | mass (kg)[b] | tag type | sampling frequency (Hz) | sensors |
|---|---|---|---|---|---|---|---|---|---|
| fish no. 1 | 18 Oct 2014 | 18:04 | 46.07112° N | 61.5799° W | 229 | 234 | CEFAS G6a | 30 | acc/depth/temperature |
| | 19 Oct 2014[c] | 12:41 | 46.00212° N | 61.6224° W | 250 | 312 | CEFAS G6a | 30 | acc/depth/temperature |
| | 25 Oct 2014 | 18:50 | 46.05933° N | 61.5606° W | 251 | 317 | CATS-Cam | 40 | acc/depth/temperature/speed/video |
| fish no. 2 | 26 Oct 2014[c] | 10:14 | 46.18592° N | 61.4595° W | 242 | 281 | CATS-Cam | 40 | acc/depth/temperature/speed/video |

[a]Curved fork length.

[b]Estimated using a length–length and length–weight conversion [56].

[c]Tag recovered.

the rear of the fish to validate signals from the accelerometers and to document the fishes' environment. The camera system was a Replay Mini action camera, custom fit into a hydrodynamic housing. The camera had a frame rate of 30 fps and a resolution of 720 p.

### 2.2.2. CEFAS G6a

G6as (CEFAS Technology, Lowestoft, USA) were encased in a positively buoyant housing and coupled with a VHF transmitter (ATS 3-stage transmitter, Insanti, MN, USA) described in detail in Whitmore *et al.* [33]. The tag package was fixed to the fin using the same materials as described for our CATS tags and was using the same GTRs as release mechanisms. Once the GTR would completely corrode, the link between the two zip-ties would break and tag would release from the fish and float to the surface to start transmitting its VHF signal for recovery. CEFAS G6a sensor array contained tri-axial accelerometers sampled at 30 Hz here, as well as temperature and depth sampled at 1 Hz.

## 2.3. Sensor data

We calibrated accelerometers of all tags to $g$ (9.81 m s$^{-2}$) according to Wilson *et al.* [57]. Acceleration data consist of two signals, one is related to the attitude of the tag in relation to gravitational field (termed static or gravitational acceleration) and the other to movement (dynamic acceleration). These signals can be estimated by various smoothing techniques to remove the gravitational component from the raw acceleration. Based on iterative visual assessment, we found that a 1.5 s second-order Sovitzky– Golay filter captured the dynamics most appropriately. This filter was applied to all three axes of the accelerometer. The surging acceleration was used to compute the pitch angle of the tuna (the angle between the longitudinal axis of the fish and the horizon), through an arcsine transformation. Since no tag can be perfectly aligned with the longitudinal axis of the animal, it is necessary to calculate an offset between the angle of the tag and that of the animal. To do so, we used the method of Kawatsu *et al.* [58] and regressed vertical velocity and pitch angle; the intercept of this relationship corresponds to this offset; i.e. when fish swim in a level fashion (VV = 0 ms$^{-1}$), the tag will have an orientation other than horizontal. This offset can simply be subtracted from the entire dataset (see also [29]).

Lateral dynamic accelerations correspond to the tail-beating of fish. To estimate locomotory effort by the fish, we calculated the tail-beat frequency from the dynamic lateral acceleration and performed a continuous wavelet transform (CWT) on these data to estimate the dominant frequencies in Ethographer [59] implemented in IgorPro (Wavemetrics, Lake Oswego, OR, USA). This yielded a time-series of tail-beat frequencies with temporal resolution of 1 s. Separately, we used a custom algorithm to identify passive glides, since our CWT missed a considerable amount of segments that were classified as active swimming, despite no clear discernible tail-beat being visible. Briefly, the lateral dynamic acceleration data were first smoothed using a second-order Sovitzky–Golay filter. The data were then grouped into consecutive 120 s bins and the root sums of squares (RSS) calculated for each bin to quantify variation in signal amplitude over time. The time-specific RSS values multiplied by a constant scaler were then used as a threshold to differentiate swimming versus gliding based on the RSS values for a 15-point sliding window. Values above the threshold were categorized as swimming, while those below were categorized as gliding. The two datasets were combined to yield a single time-series; any time step defined as passive in our glide code was used to overwrite the corresponding frequency in the CWT dataset.

### 2.3.1. Measurement of speed

Speeds were calculated from the number of revolutions recorded by a small paddle-wheel attached to the outside of the tag. The paddle-wheel contained two small magnets of opposite polarity, affixed to two opposite paddles. Rotation in the paddle-wheel resulted in small oscillations in the data of the tags magnetometer, quantified using a CWT in the Ethographer package [60]. These data yielded a separate dataset of paddle-wheel revolutions per second, which was used in the calibration procedure. To estimate swimming speeds, we used the measurements of depth and pitch of the diving tuna to estimate a swimming speed. Given that pressure sensors provide very accurate data, one can calculate a velocity when the animal is swimming vertically (pitch = 90°); however, this is rarely the case. Instead, we used trigonometry to calculate a velocity based on the pitch and vertical velocity [61]. Estimates from trigonometry will have increasingly large errors (approaching infinity at pitch = 0) at shallow angles, we therefore restricted the calibration procedure to sections of data where the angle was greater than 20°. We found that paddle-wheel rotations increased with our estimated swimming speed, but we

found some variation in the relationship, which is expected based on error in the estimation of speed from pitch. Based on binned rotation/estimated velocity plots, it became apparent that a nonlinear fit was required to capture dynamics appropriately, especially at low speeds. We used a fourth-order polynomial for the calibration, which explained 61% of the variation of the relationship (99% for the binned data). Simpler functions were not able to capture the dynamics at slow speeds and would have resulted in an overestimation of velocity. We subsequently applied the calibration over the entire dataset. We did not record any slow-speed stalling in the sensor, but reached a maximum resolvable speed at approximately 3.25 m s$^{-1}$ (see electronic supplementary material, figure S2). This was due to paddle-wheel rotations reaching 20 Hz at this velocity, equivalent to the Nyquist frequency of the sampling regime in the magnetometer (40 Hz).

# 3. Results

## 3.1. Deployment and recovery of accelerometry tags

Four biologging tags were deployed to obtain the first recordings of speed and accelerometry of wild Atlantic bluefin tuna (T. thynnus); two CATS camera tags and two CEFAS G6as (figure 1 and table 1). Two tags were not recovered. The CATS camera tags did not release at the expected pop-off time. Both tags came up approximately 60 days from the expected time of release, approximately 100 miles off the coast of Nova Scotia, close to Sable Island (electronic supplementary material, figure S1). One of the tags was recovered by boat and crew with a hand-held ARGOS receiver. The second CATS tag also popped up in the same vicinity, but due to severe weather and the distance from shore, it could not be collected before the battery of the PTT ran out and transmission ceased. The reason for the delayed pop-off is not known, but we speculate that the stiff zip-ties got jammed in the grooves of the tag and the fin and only working themselves loose following an extended period at liberty. The G6a containing only a VHF transmitter was found by a member of the public near the causeway between Cape Breton Island and mainland Nova Scotia in the following spring. The initial scanning of the data revealed that the G6a popped off after 72 h (hereafter referred to as fish no. 1), whereas the CATS diary (hereafter referred to as fish no. 2) recorded while attached to the fish for 48 h, until the batteries of the data-loggers were exhausted. Together, these data provided 5 days of kinematic data on Atlantic adult bluefin tuna swimming freely with a bathymetric shelf limitation.

## 3.2. Behaviour following capture

Atlantic bluefin tuna were brought on to the deck for electronic tagging and released head first from the vessel following the deployment procedures using established techniques. In both cases, after release, the fish dove and exhibited a gliding descent (no tail-beating) to depths of 20–30 m. Both fish displayed strong tail-beating activity with higher tail-beat frequency of greater than 1.5 Hz for several hours (see figure 2) followed by a gradual decrease towards a more sustained level of activity of less than 1 Hz after approximately 6 h. In the single fish (fish no. 2) for which swim speed was directly measured, the data were characterized by a post-release response of fast swimming speeds of approximately 2.5 m s$^{-1}$ (approx. 1 body lengths s$^{-1}$), followed by a reduction to a routine speed of greater than 1.5 m s$^{-1}$ (approx. 0.75 body lengths s$^{-1}$) over the course of 6 h. Tuna performed regular DVMs in a bathymetrically limited environment (approx. 30–40 m); fish no. 1 occurred at depths around 30–40 m during the day and spent the night primarily near the surface with occasional dives to 30–35 m and encountered temperatures ranging from 12 to 14°C. Fish no. 2 also performed a DVM while encountering temperatures of 8–12°C (figure 2). Unlike the DVMs associated with diving, the fish displayed contrasting diel patterns in their locomotory activity, with fish no. 1 primarily increasing activity during night-time and fish no. 2 during daytime (figure 2).

## 3.3. Kinematics and swimming performance during horizontal locomotion

During episodes of horizontal swimming, defined as tuna having a mean absolute vertical velocity less than 0.1 m s$^{-1}$ and not undertaking vertical movements greater than 10 m, swimming speed showed a linear relationship with tail-beat frequency for fish no. 2 (figure 3). Both tuna showed largely skewed frequency distributions of tail-beat frequency, with the most used frequencies being close to the lowest recorded; the modal frequencies were 0.80 and 0.95 Hz for fish no. 1 and no. 2, respectively. In both

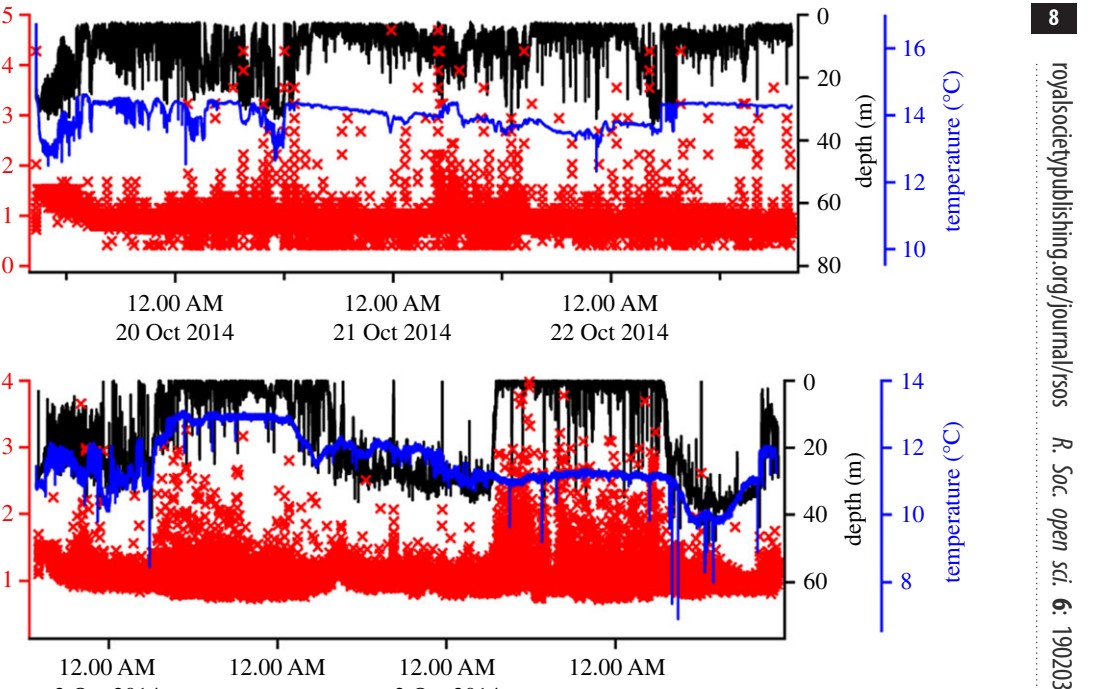

**Figure 2.** Overview of the DVMs and patterns in activity of the two giant Atlantic bluefin tuna tagged in the Gulf of St Lawrence, Canada.

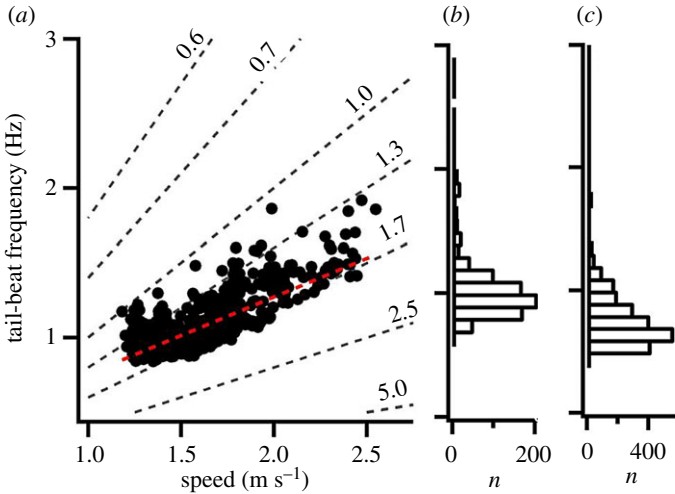

**Figure 3.** Tail-beat frequencies during horizontal swimming of Atlantic bluefin tuna. Relationship between tail-beat frequency and swimming speed for level swimming in fish no. 1, red stippled line shows the linear fit ($r^2 = 0.59$) (a) and the respective frequency distributions of tail-beat frequencies for the same fish (b) and fish no. 2 (c) during level swimming for the entire duration. Revealing that fish spent the vast majority at tail-beat frequencies below 1 Hz, equivalent to a swimming speed of less than 1.75 m s$^{-1}$ or less than 0.7 body lengths s$^{-1}$. Stippled lines represent constant stride-lengths in metres.

tuna, sustained tail-beat frequencies approaching 2 Hz were present but were only rarely observed during sustained swimming (defined here as the mean tail-beat frequency over a 2 min interval) and were largely restricted to the beginning of the deployments when fish were probably stressed, although isolated records of such high tail-beat frequency were present later on.

Importantly, a main observation of our kinematic data from wild fish are that the majority of the time, both tuna did not display continuous tail-beating, but displayed a mixture of continuous and intermittent locomotion (figure 4), as ascertained from the accelerometers and animal-attached video recordings of

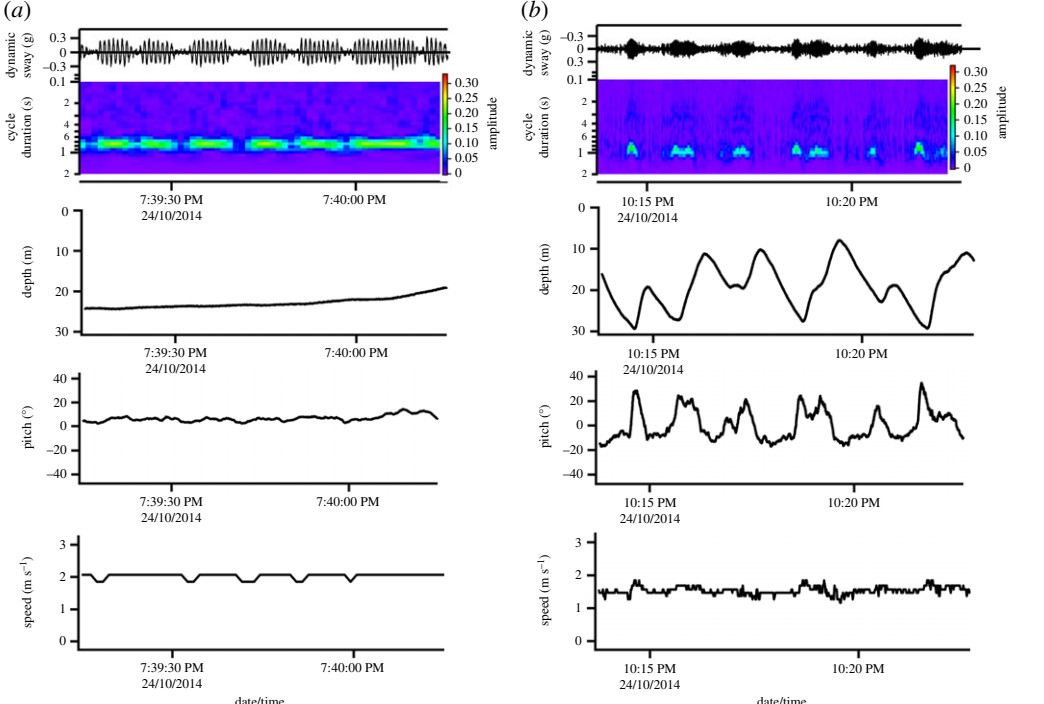

**Figure 4.** Patterns of intermittent locomotion observed in giant Atlantic bluefin tuna. (*a*) Classic burst–coast swimming [18], where vigorous fast swimming is interspersed with brief periods of gliding (also see electronic supplementary material) at near horizontal body-orientation. (*b*) Classic two-stage locomotion as proposed by Weihs [17] for negatively buoyant fish. Vertical movements are performed at relatively low speeds and intermittent locomotion primarily arises from prolonged periods of gliding during shallow (approx. 20°) descents and shallow (approx. 15°) ascents comprised active swimming. Please see supplementary video files for examples of the different gait patterns employed by this fish.

**Table 2.** Summary diving statistics for two Atlantic bluefin tuna equipped with data-loggers.

| fishID | | *n* | duration (s) | depth (m) | speed (m s$^{-1}$) | pitch (°) | % active swimming |
|---|---|---|---|---|---|---|---|
| fish no. 1 | ascent | 115 | 36.09 ± 12.39 | 15.18 ± 4.18 | n.a. | 18.09 ± 6.10 | 98.47 ± 3.38 |
| | descent | 90 | 50.01 ± 19.10 | 13.78 ± 4.55 | n.a. | −9.65 ± 5.86 | 44.24 ± 28.88 |
| fish no. 2 | ascent | 334 | 25.00 ± 11.29 | 9.95 ± 5.40 | 1.75 ± 0.25 | 14.97 ± 4.91 | 90.37 ± 0.09 |
| | descent | 363 | 33.82 ± 18.88 | 20.06 ± 10.48 | 1.60 ± 0.30 | −9.61 ± 4.33 | 30.34 ± 23.09 |

fish no. 2. At the routine speed of less than 1.5 m s$^{-1}$, fish no. 2 largely swam continuously, although brief periods of gliding were still noted, but these did not appear as stereotypical as the intermittent locomotion following release from the tagging vessel or those during vertical movements.

## 3.4. Kinematics and swimming performance during vertical movements

Tuna performed 'large'-scale vertical movements (dives), defined here as depth movements with a continuous vertical velocity exceeding 0.1 m s$^{-1}$ over a minimum change in depth of 10 m. Due to the shallow nature of the Gulf of St Lawrence (less than 50 m in the region where the fish were released), these dives in the first 48 h post-release are probably bathymetrically limited, as evidenced in the video record of the fish carrying a video camera. Dives were typically asymmetrical in shape; tuna descents were characterized by shallower pitch angles around approximately −9°, whereas ascents were more typical of approximately 17° (table 2). Descents were primarily composed of glides and only approximately 30–40% active swimming events and tail-beat frequencies between 1 and 2 Hz. Gliding was almost absent in ascents, with ascents being composed of greater than 90% active swimming and tail-beat frequencies of 1–2 Hz, depending on swimming speed (table 2; electronic

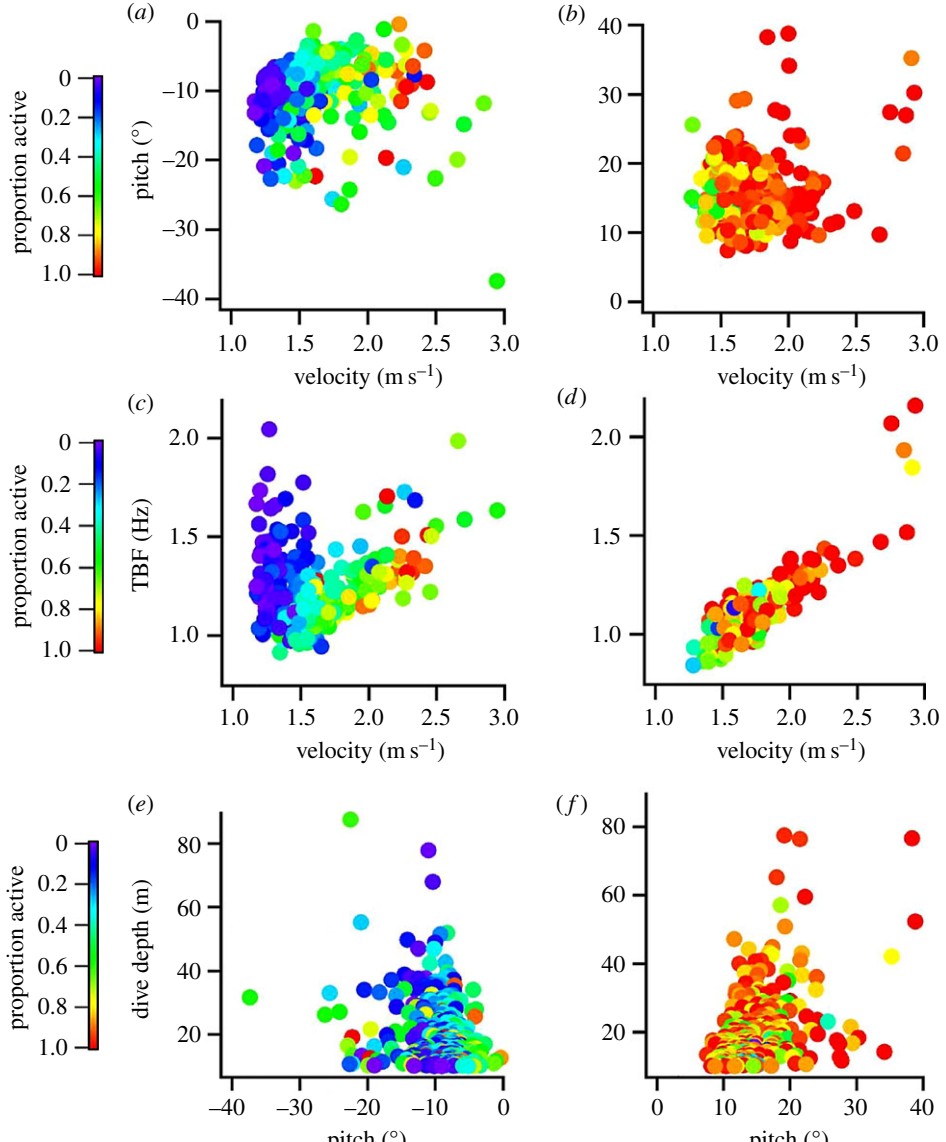

**Figure 5.** Kinematic parameters for an Atlantic bluefin tuna during prolonged (greater than 10 m) descent (*a,c,e*) and ascents (*b,d,f*). Poor relationships were found for the majority of kinematic parameters (table 3), with the exception of swimming speed, tail-beat frequency (TBF) and the %time engaged in active swimming (proportion active in legend) showing moderate to strong relationships.

supplementary material). Fish no. 2, for which swim speeds were available, travelled at a mean speed 1.6 m s$^{-1}$ during descents and 1.75 m s$^{-1}$ during ascents (table 2).

Pearson's product moment correlations revealed that tail-beat frequency was tightly correlated with swim speed during ascents but to a lesser extent in descents (figure 5 and table 3). The proportion of time spent actively swimming, as opposed to gliding, showed the opposite trend, with swim speed and percentage time actively tail-beating being tightly positively correlated for descents and to a lesser degree in ascents (table 3). Dive depths displayed poor to no correlations with other parameters, apart from a modest positive correlation with tail-beat frequency during descents (table 3). Dive angles also showed moderate correlations with tail-beat frequency during ascents, but not descents (table 3).

Fish no. 2 with a direct swimming speed sensor showed that tail-beat frequency was tightly correlated with swimming speed in both ascents and descents (ANCOVA, $F_{2.623} = 594.3$, $r^2 = 0.67$, $p < 0.0001$), with ascent tail-beat frequency marginally higher at a given swim speed ($p < 0.0001$; figure 6), with no significant interaction ($p = 0.11$). However, descents featured a larger proportion of glides (table 2). To incorporate the difference in the amount of gliding between ascents and descents, we calculated an additional metric of swimming performance, the rate of tail-beating, as the mean number of tail-beats per second to incorporate gliding into our estimates of overall locomotor activity. This revealed

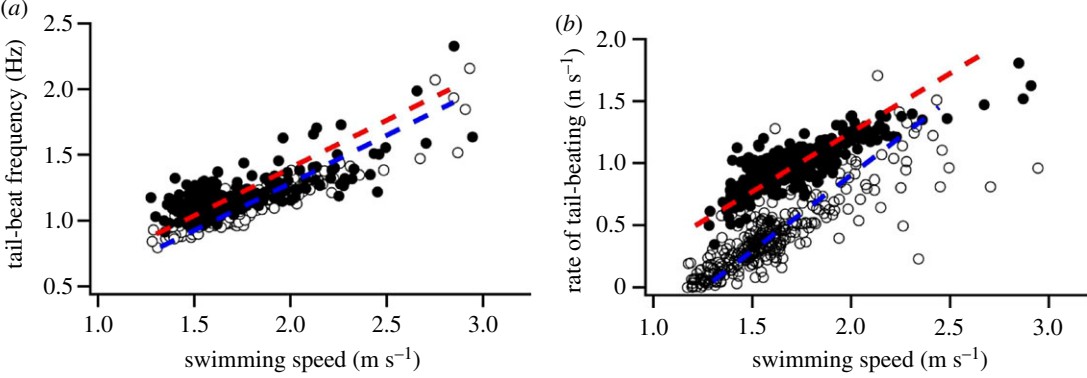

**Figure 6.** Relationship between tail-beat frequency and swimming speed of diving Atlantic bluefin tuna. (a) When actively tail-beating (number of tail-beats/time actively tail-beating), frequencies during ascent (filled circle, red dashed line) were marginally elevated compared with tail-beating during descent (open circle, blue dashed line). (b) When incorporating the proportion of time fish did not beat their tail and actively glided, the rate of tail-beating (number of tail-beats/(time actively tail-beating + time passively gliding)) is significantly reduced in descents compared to ascents, approaching zero tail-beats in slow descents with swimming speeds of $1.25 \text{ m s}^{-1}$. At greater speeds, the difference in locomotory effort between ascents and descents increasingly subsides.

**Table 3.** Pearson's product moment correlations between five kinematic parameters of Atlantic bluefin tuna during vertical excursions (greater than 10 m). Data for 448 ascents and 452 descents. Each pair of values represents the Pearson's $r$ for ascents (top value) and descents (bottom value) of the paired parameters. Significant correlations are indicated by the asterisk. Note that only one tuna carried a speed sensor and subsequently fewer dives were available for speed correlations (334 ascents and 363 descent).

| $r$ ascent $r$ descent | tail-beat frequency | prop. active | dive depth | dive angle |
|---|---|---|---|---|
| swim speed[a] | 0.81*** | 0.41*** | 0.07 | 0.15* |
| | 0.31*** | 0.77*** | 0.01 | 0.01 |
| tail-beat frequency | | 0.32*** | 0.12* | 0.44*** |
| | | 0.1* | 0.31*** | −0.05 |
| prop. active | | | 0.08 | 0.16** |
| | | | −0.15** | 0.27** |
| dive depth | | | | 0.27*** |
| | | | | −0.16** |

*$p < 0.05$; **$p < 0.005$; ***$p < 0.0005$.
[a]Data only available for one fish.

drastically lower activity for descents (ANCOVA, $F_{3,621} = 592.1$, $r^2 = 0.78$, $p < 0.0001$), especially at lower speeds, which were primarily composed of glides (figure 6). The difference between ascending and descending subsided at faster speeds, with a significant interaction between tail-beat, swimming speed and whether the fish was ascending or descending ($p < 0.0001$, figure 6). Although lacking the swimming speed data, bluefin no. 2 showed qualitatively similar patterns with respect to its gliding behaviour during ascents and descents.

## 4. Discussion

Biologging tags are rapidly improving our capacity to quantify many aspects of the biology of animals occupying the most inaccessible places of our planet. Electronic tags have historically allowed the elucidation of the spatio-temporal patterns in the behaviour in diverse taxa, including tuna, such as seasonal migrations or diel changes in behaviour and habitat use [10]. The advancing sensory

capabilities and increased memory of biologging tags are now offering similar revolutionary insights into the fields of comparative physiology and biomechanics of even the most enigmatic animals in the wild [60,61,62]. Bluefin tuna are extreme athletes with the capacity to rapidly travel across ocean basins, with remarkably quick migrations from low-latitude spawning grounds to high-latitude foraging grounds [63], underpinned by unique physiological and biomechanical adaptations [1]. While archival tag records have revealed the remarkable depths to which these fish can dive and their use of the water column, this study provides a first effort to examine the swimming kinematics in wild tuna. The use of a high-resolution biologging tag coupled with a video device has revealed that in addition to morphological specializations for high-performance swimming, Atlantic bluefin tuna also showed considerable repertoire of swimming patterns that are hypothesized to improve their locomotor performance.

## 4.1. Intermittent locomotion and locomotor performance

Various types of intermittent locomotion have been hypothesized to increase the economy of movement in fishes—here, we provide empirical evidence for earlier theoretical work and confirm what has been posited for decades to occur in free-ranging tuna [17,18]. Firstly, Atlantic bluefin tuna use their negative buoyancy, and lifting surfaces provided by their fins, to glide for prolonged durations during descent and do so at very shallow angles, thus translating vertical forces into horizontal distance. This represents an important aspect of Weihs' two-stage locomotion model [17,64], suggesting that some dives could be performed to increase the economy of horizontal movements, which is in broad agreement with captive work on bluefin tuna [54,55]. Low swimming speeds during those dives are most likely also crucial for increasing efficiency, since descending at faster speeds appears to require the fish to spend more time actively swimming, as hydrostatic forces of negative buoyancy are not sufficient to overcome drag at those faster speeds. Remarkably, much of the video once the fish was calm following release revealed that bluefin tuna only used minor locomotor activity, despite still cruising at appreciable speeds (approx. $1.5 \, \mathrm{m \, s^{-1}}$). Conservation of muscle activity is clear in these long sequences of gliding with intermittent tail-beats (see supplementary video). Dives also feature marginally asymmetrical geometries, ascents are both faster and steeper than descents, thus decreasing the time engaged in the energetically expensive portion of a dive and increasing the amount of time engaged in gliding phases at shallow angles, thus increasing horizontal distance traversed per unit energy. Similar geometries have been noted by Holland *et al.* [50] for big-eye tuna calculated from active acoustic tracking of fish with depth-sensitive tags, which only provided modest savings based on the observed trajectories, which were very shallow at approximately $6°$ for ascents and descents, compared to the steeper diving geometries of approximately $10–15°$ recorded for bluefin here. Geometries calculated from active acoustic tracking, however, suffer from errors that have not been accounted for, namely inaccuracy in the location of fish and not accounting for ocean currents that would alter the calculated angles. Calculations of pitch angles from accelerometers do not suffer from the same biases.

Importantly, intermittent locomotion was not constrained to descents during vertical movements, but also featured in horizontal locomotion. However, it was largely reserved to the beginning of the deployment, whereas the latter part of the record featured more continuous locomotion at comparable fast swimming speeds. However, such fast swimming speeds were only present sporadically later on in the record. It is well established that fish switch from continuous swimming gaits to burst–coast gaits when aerobic muscle contractile velocities are insufficient to achieve efficient swimming at high speeds [18,65]. More records of longer duration are required to better understand the kinematics of tuna, especially during higher swim speeds that rarely were recorded in the short duration deployments.

## 4.2. Diving kinematics

The novel tagging data provided from the accelerometry and video biologging tags provide new information on how bluefin tuna are able to modulate swimming speed during diving and begin to answer some long-standing questions about the biomechanics of these fish in the wild. From these initial data, we have learned that the Atlantic bluefin tuna alter both the frequency of their tail-beat and the amount of time that is spent passively gliding. In gravity-assisted locomotion, i.e. when tuna descend due to their negative buoyancy [66], speed is primarily modulated by the amount of time spent passively gliding, especially at lower speeds, whereas at higher speeds (greater than $1.75 \, \mathrm{m \, s^{-1}}$), tail-beat frequency is kinematically more important as the time fish spend gliding reduces to approximately

40%. Swimming speed during ascents is largely determined by tail-beat frequency and to a lesser extent by the amount of gliding. The clear dichotomy between the modulation of speed between ascents stems from two basic principles; negative buoyancy of the bluefin tuna is translated into forward motion through lift production of the pectoral fins, allowing a tuna to glide for prolonged durations [67]. On ascent, the bluefin tuna has to overcome this negative buoyancy, meaning more thrust has to be generated. To our knowledge, no data on the buoyancy of any bluefin tuna are available; however, based on the data of other true tunas, such as big-eye and yellowfin, featuring equally small gas bladder and dense tissue [66,67], bluefin tuna are negatively buoyant (B.A.B. 2014, personal observation). Larger amounts of passive gliding during descents for the tuna studied here support this [29,31].

The comparison of the kinematics between the two phases of diving (ascent and descent) revealed interesting patterns; the relatively minor differences in the tail-beat frequency of tuna during ascents and descents at a given swimming speed were surprising. At first hand, this may suggest that overcoming buoyant forces (i.e. change in potential energy) represents a minor component of the power required to swim. However, tuna glide significantly more when buoyancy assisted (descending) than buoyancy impeded (ascending). When incorporating the per cent time spent gliding into our approximation of locomotor effort, descent with gravity assistance and ascent with gravity hindrance show clear differences. Importantly, the difference in locomotor activity between ascending and descending swimming was found to decline with increasing swimming speed. This is an expected pattern based on first principles; the induced drag of a lift-producing surface, such as the pectoral fin of a tuna [66] is inversely related to the square of swimming speed, whereas drag of the fishes body (parasite drag) is proportional to the squared swimming speed [68,69]. Thus, at faster speeds, the relative cost of counteracting negative buoyancy becomes increasingly negligible compared to the costs related to parasite drag; subsequently, at the highest speeds, we would expect there to be little difference between the energetic costs of ascents and descents beyond the difference in potential energy. Moreover, faster speeds also incur more hydrodynamic resistance, and therefore, fish have to increase the proportion of time actively swimming as opposed to gliding. Therefore, relative reduction in mechanical work by repeated diving decreases as swimming speeds increase, as demonstrated in Weihs' original theory [17].

The similarity between tail-beat frequency during ascent and descent (before accounting for per cent gliding), however, may represent more than an artefact, as it suggests that swimming speed, not buoyancy represents the key parameter governing tail-beat frequency. The reason for this may be related to the gearing of the muscle and hydrodynamic properties of the tail [70,71]. The hydrodynamic properties of the musculoskeletal and hydrodynamic systems are probably optimized to provide optimum thrust at a given swim speed and moving outside of this range may result in a loss of efficiency, similar to riding a bike in the wrong gear. By interspersing glides with active tail-beating, tuna effectively extend this gearing ratio, rather than using tail-beat frequencies that may involve lower efficiency when less thrust is required. Similar theory has been shown to apply to intermittent flapping flight in birds [72]. Of course, fish can alter other kinematic parameters that accelerometers cannot easily allude too, namely tail-beat amplitude or the stiffening of the caudal region, which may be altered in response to changes in their hydrodynamic environment. Changes in amplitude may explain the marginal difference in tail-beat frequency between ascents and descents here.

It is worth noting, that our discussions related to vertical movements providing an energy saving for fish rely on an important assumption in Weihs' theory [17], namely that the efficiency of transforming chemical energy into mechanical work must differ between ascent and descent [27]. Weihs originally proposed that the drag incurred by a fish swimming with lateral undulations should be in the region of two to three times greater than those of a rigid, gliding fish. Therefore, if a fish can propel itself passively using gravity during descent, it would incur substantially less drag than if it were to swim continuously. Takagi et al. [54], using computational fluid dynamics, quantified and confirmed the drag coefficient of juvenile Pacific bluefin tuna to be greater during undulatory swimming than gliding by a factor of two, further supporting that tuna may save energy through repeated diving. Some caution needs to be exercised in translating these data to other fishes, as this effect may not be universal and might depend on the kinematics of the species in question.

Our analysis has considered the forces and kinematic responses of tuna to be somewhat constant during ascent and descent. This, however, is not strictly true; despite the gas bladders of bluefin tunas being small for their body size [67], it provides an appreciable amount of hydrostatic force. Especially for fish performing vertical movements, these hydrostatic forces change with depth, due to increasing ambient pressure and a reduction in gas bladder volume following Boyle's Law. We found some evidence that the depth of dives (and by extension the amount of hydrostatic forces provided by the

bladder) influences the diving of these fish. Tuna ascending from deeper depths beat their tails faster compared to shallower dives; however, they also dove at a steeper angle, making it difficult to distinguish between the influence of the changing hydrostatic forces or the changing diving geometry. We did not observe the drastic changes in locomotor effort that have been observed in other species with larger bladders, such as sturgeon or catfish [73,74]. This may be a result of the changes in hydrostatic forces during vertical movement in tuna being somewhat smaller than those species that have large gas bladders and near-neutral buoyancy at their depth of residence [67].

Our discussion has so far focused on the energetic consequences of diving behaviour; however, this should not necessarily imply that these tuna dove for the purpose of improving their locomotor performance, but rather in a manner that improved their locomotor performance. Fish have been shown to dive for a variety of reasons and that the functions of these vertical excursions are not mutually exclusive and can change daily [30,44,75]. Certainly for Atlantic bluefin tuna and other species of tunas, thermoregulation has been a prominent explanation for vertical excursions, permitting fish to feed below the thermocline while maintaining elevated muscle temperatures [48,49]; big-eye tuna diving for thermoregulatory purposes generally feature very high vertical velocities (approx. $1–2\,\mathrm{m\,s^{-1}}$) [48], which suggest fish dive steeper than Atlantic bluefin tuna in our study. Indeed, the relatively shallow water of the Gulf of St Lawrence is well mixed and offers little opportunity for thermoregulation. Instead, tuna probably perform repeated vertical movements for the purpose of locating their schooling prey. It is generally thought that Atlantic bluefin tuna congregate annually in the Gulf of St Lawrence to feed on Atlantic herring (*Clupea harengus*) and Atlantic mackerel (*Scomber scombrus*) and anecdotal evidence from commercial fishermen suggest that fish gain substantial mass during their summer residence (A.C.G. and R.J.S. 2014, personal communication). Future research should deploy similar tags in areas where the fishes' diving behaviour are constrained by environmental conditions, such as very low temperatures below the thermocline or vertical stratification in oxygen concentration, since diving kinematics may be drastically altered by changes in temperature and/or oxygen availability.

Our work demonstrates that tuna use a number of strategies that are consistent with theoretical models of movement economy, confirming long-standing hypotheses about the manner in which free-ranging tuna move. These insights largely stem from our methodological advancements in the external attachment of data-loggers to tuna using their small, but rigid dorsal fin and a streamlined tag design. Our results lend further support to theory of strong convergence in the evolution of swimming kinematics across all fluid media; where patterns of intermittent locomotion coupled with vertical movements have been demonstrated in a range of distantly related clades, such as aves, phocids, mysticetes and elasmobranch fishes [24,72,76]. This suggests that the physical forces acting on moving animals are under strong selective pressure and result in the widespread evolution of behaviours and morphologies that reduce unnecessary energy expenditure [24,32].

Ethics. Tuna were tagged under permit from the Department of Fisheries and Ocean Gulf Region, Canada (SG-GNS-14-132A). All procedures were approved by Stanford University's Institutional Animal Care and Use Committee (13865).
Data accessibility. Data available from the Dryad Digital Repository: https://doi.org/10.5061/dryad.7vp288f [77].
Authors' contributions. A.C.G. and B.A.B. conceived the study. A.C.G., R.J.S. and S.G.W. performed the fieldwork. A.C.G. and J.J.D. analysed the data. A.C.G. drafted the manuscript with all authors contributing and approving the final version.
Competing interests. We declare we have no competing interests.
Funding. Work was funded by the Office of Naval Research.
Acknowledgements. We would like to thank Gil Iosilevskii, Denise Greig and an anonymous referee whose comments greatly improved this manuscript. We would also like to thank Dennis Cameron, Lloyd MacInnes, Duncan Sutherland, Ceilidh Fishermen Co-Op, Department of Fisheries and Oceans, Dr Michael Stokesbury and the community of Port Hood for supporting our work. We would like to thank Nicholas Whitney for providing floats for G6as and Mike Castleton for producing electronic supplementary material, figure S1. We thank TAG A Giant of the Ocean Foundation for support.

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
