## [Reviewer comments · Royal Society Open Science]

Review History

RSOS-180946.R0 (Original submission)

Review form: Reviewer 1

Is the manuscript scientifically sound in its present form?

Yes

Are the interpretations and conclusions justified by the results?

Yes

Is the language acceptable?

Yes

Is it clear how to access all supporting data?

Yes

Do you have any ethical concerns with this paper?

No

Have you any concerns about statistical analyses in this paper?

No

Recommendation?

Accept with minor revision (please list in comments)

Comments to the Author(s)

The authors deployed accelerometers and speed sensors on large Atlantic blue fin tunas to show their fine-scale swimming behavior in the wild. Although sample size is only two, I think the authors did a great job, given the difficulties in obtaining this kind of data from large, wild tunas (tag recovery points are very far). The data are new to science, and worthy of publication. I am especially impressed by fast cruising speed of tunas. My comments are minor, and I believe this paper will be a great contribution to Royal Society Open Science.

L43 Suggest deleting "a number of".

L53 Delete comma after "Tunas".

L77 "Up to".

L159-165 Suggest moving this part to methods

L168 CFL should be spelled out.

L184-185 Suggest moving this sentence to methods

L227 Suggest changing "dive angles" to "pitch angles" to avoid misunderstanding.

L294-295 Here I cannot catch the point. Why are fast and steep ascents related to increasing horizontal distance? What is "per unit of energetically intensive ascent"?

L326 "between ascents and descents"

L391 Thermoregulation is functionally related to foraging in those fishes.

L406 Suggest deleting "a number of".

L525-549 It's unclear to me why you didn't calibrate your paddle-whales in flow tanks (or something similar)? I am concerned that the pitch method doesn't work well for fishes due to shallow pitch angles in general (and that's why the relationships are so noisy?).

Review form: Reviewer 2**Is the manuscript scientifically sound in its present form?**

No

Are the interpretations and conclusions justified by the results?

No

Is the language acceptable?

Yes

Is it clear how to access all supporting data?

No

Do you have any ethical concerns with this paper?

No

Have you any concerns about statistical analyses in this paper?

No

Recommendation?

Reject

Comments to the Author(s)

Attached (Appendix A).

Decision letter (RSOS-180946.R0)

03-Sep-2018

Dear Dr Gleiss:

Manuscript ID RSOS-180946 entitled "Direct Measurement of Swimming and Diving Kinematics of Giant Atlantic Bluefin Tuna (*Thunnus thynnus*)" which you submitted to Royal Society Open Science, has been reviewed. The comments from reviewers are included at the bottom of this letter.

In view of the criticisms of the reviewers, the manuscript has been rejected in its current form. However, a new manuscript may be submitted which takes into consideration these comments.

Please note that resubmitting your manuscript does not guarantee eventual acceptance, and that your resubmission will be subject to peer review before a decision is made.

Your resubmitted manuscript should be submitted by 03-Mar-2019. If you are unable to submit by this date please contact the Editorial Office.

Please note that Royal Society Open Science will introduce article processing charges for all new submissions received from 1 January 2018. Charges will also apply to papers transferred to Royal Society Open Science from other Royal Society Publishing journals, as well as papers submitted as part of our collaboration with the Royal Society of Chemistry

(<http://rsos.royalsocietypublishing.org/chemistry>). If your manuscript is submitted and accepted for publication after 1 Jan 2018, you will be asked to pay the article processing charge, unless you request a waiver and this is approved by Royal Society Publishing. You can find out more about the charges at <http://rsos.royalsocietypublishing.org/page/charges>. Should you have any queries, please contact openscience@royalsociety.org.

on behalf of Dr Denise Greig (Associate Editor) and Prof. Kevin Padian (Subject Editor)
openscience@royalsociety.org

Subject Editor Comments:

We hope that the authors will be able to address the comments, recalibrate as necessary, and resubmit having addressed all reviewers' comments. Thanks for submitting.

Associate Editor Comments to Author (Dr Denise Greig):

These are fascinating data and it is clear that a lot of work went into this project. Deploying new biologging technology onto fast moving pelagic species is no small feat, however the methods and analyses need to be revised for clarity. Additionally one reviewer had serious concerns about the post-processing of the telemetry data. I hope you will be willing to address his in-depth concerns and resubmit this manuscript to RSOS.

My specific questions/comments below:

Great introduction!

Methods

The tag descriptions are confusing... I did not see anything about camera portion of the CATS tag when sat tag portion and casing were described (although it is pictured in Figure 1). What about the depth and temperature sensors - where on the tag are they located? Are they part of the Wildlife Computers and ATS portions of the tag? comparable between the two tag types? Programmed the same way?

Do you expect positively buoyant tag casings to affect your kinematic measurements?

Line 441 please change parenthetical to (see Figure 1 for CATS tag and Whitmore et al for CEFAS G6a)

Results - the results are confusing/missing some information. Table 2 is helpful for the methods and results, but in the text it is difficult to figure out which data were acquired from which fish. I think maybe having the Methods at the end adds to the confusion - is there a reason that you chose to do this?

Line 160 - first mention of the CATS tag (please add abbreviation) -

Unclear how "glides" are defined (using the video)? Video mentioned in the Discussion but not Results

Line 234 - this is fish #1 also correct?

Line 453 and 459 - two undefined abbreviations (ABFT and GTR), GTR is defined later in Line 466

Line 459 - please add a references for the washers that are "previously described"

Figure 1D - is the photo upside down?

Figure 3..I thought you only had data from Fish #1 (CATS) and Fish #2 (G6a)? This figure shows data from Fish #4?

Reviewers' Comments to Author:

Reviewer: 1

Comments to the Author(s)

The authors deployed accelerometers and speed sensors on large Atlantic blue fin tunas to show their fine-scale swimming behavior in the wild. Although sample size is only two, I think the authors did a great job, given the difficulties in obtaining this kind of data from large, wild tunas (tag recovery points are very far). The data are new to science, and worthy of publication. I am especially impressed by fast cruising speed of tunas. My comments are minor, and I believe this paper will be a great contribution to Royal Society Open Science.

L43 Suggest deleting "a number of".

L53 Delete comma after "Tunas".

L77 "Up to".

L159-165 Suggest moving this part to methods

L168 CFL should be spelled out.

L184-185 Suggest moving this sentence to methods

L227 Suggest changing "dive angles" to "pitch angles" to avoid misunderstanding.

L294-295 Here I cannot catch the point. Why are fast and steep ascents related to increasing horizontal distance? What is "per unit of energetically intensive ascent"?

L326 "between ascents and descents"

L391 Thermoregulation is functionally related to foraging in those fishes.

L406 Suggest deleting "a number of".

L525-549 It's unclear to me why you didn't calibrate your paddle-whales in flow tanks (or something similar)? I am concerned that the pitch method doesn't work well for fishes due to shallow pitch angles in general (and that's why the relationships are so noisy?).

Reviewer: 2

Comments to the Author(s)

Attached

Author's Response to Decision Letter for (RSOS-180946.R0)

See Appendix B.

RSOS-190203.R0

Review form: Reviewer 2

Is the manuscript scientifically sound in its present form?

Yes

Are the interpretations and conclusions justified by the results?

Yes

Is the language acceptable?

Yes

Is it clear how to access all supporting data?

Yes

Do you have any ethical concerns with this paper?

No

Have you any concerns about statistical analyses in this paper?

I do not feel qualified to assess the statistics

Recommendation?

Accept with minor revision (please list in comments)

Comments to the Author(s)

I am glad that the authors rectified the speed data. Nonetheless, the paper needs a careful proof-reading, as 'old' (wrong) numbers have not been replaced throughout. I have marked a few places, but there can be more. Other than that, I recommend publication. It is a unique set of data.
Manuscript

1. L. 327-328: "swimming speed showed a non-linear relationship with tail-beat frequency for Fish #2 (Figure 3)."

2. L. 820. Caption to Figure 3. The speed (and stride length) were not updated

3. L. 903. Caption to Figure 6. The speed (2.5 m/s) was not updated

Supplementary

1. Figure S2 (Supplementary S2) should be replaced. It is the old one.

Data in Dryad

1. The abstract seems to reflect the 'old' speed calibration: to sustain 3 m/s, a tuna needs more than 1Hz.

2. The CATS descent data seems to reflect the old calibration. Please check.

3. Some of the columns need units and need explanation. For example, in CATS data: what are D_{min} and D_{max} ? What is the difference between TBF and TBF_Hz? (The latter is the reciprocal of the former, and hence the first TBF is probably the tail-beat period in seconds, and not the frequency). Is TBF in the second file the frequency or the period?

4. I would be happier to see representative time series (at least one full period) of yo-yo, slow speed burst-and-coast, high speed burst-and-coast, and steady swimming, rather than the averaged data appearing in Dryad (it can be down-sampled to 10 HZ with hardly any harm).

Having both would have been even better.

Miscellaneous comments

1. L. 282: "Velocity"

2. L. 294: "Cats"

3. L. 778 (Ref. [77]): I am not sure if this reference will pass the production people.

4. Figure 3: I would have added lines of constant stride length v/TBF for a reference.
5. I would have been more careful in interpretations of yo-yo and burst-and-coast strategies. Both can reduce the cost of transport only if (chemo-mechanical) propulsion efficiency is either thrust- or speed- dependent (Ref. [28], [65] and also the reference cited in footnote 2 of the first review). With constant propulsion efficiency, the swimming strategy that minimizes the cost of transport is constant-speed-constant-depth. In the context of these strategies, a few specific comments:
- L. 406-410. This is a conjecture. I am not sure if the statement is supported by the data.
- L. 435: During acceleration/ascent the fish 'pays' for the lack of activity during deceleration/descent by increasing its TBF, and therefore the frequency argument should probably be reworded. Adding the word 'red' before the word 'muscles' could solve most of the problem [65].
- L. 473-476: The difference in energetic costs of ascent and descent is independent of speed. It is just the difference in the potential energy (submerged weight times the change in depth) divided by the propulsion efficiency.
- L. 493: And also the flex at the caudal peduncle.

Decision letter (RSOS-190203.R0)

07-Mar-2019

Dear Dr Gleiss

On behalf of the Editor, I am pleased to inform you that your Manuscript RSOS-190203 entitled "Direct Measurement of Swimming and Diving Kinematics of Giant Atlantic Bluefin Tuna (*Thunnus thynnus*)" has been accepted for publication in Royal Society Open Science subject to minor revision in accordance with the referee suggestions. Please find the referees' comments at the end of this email.

The reviewers and Subject Editor have recommended publication, but also suggest some minor revisions to your manuscript. Therefore, I invite you to respond to the comments and revise your manuscript.

- Ethics statement

- Data accessibility

<http://datadryad.org/submit?journalID=RSOS&manu=RSOS-190203>

- **Competing interests**

- **Authors' contributions**

- **Acknowledgements**

- **Funding statement**

Because the schedule for publication is very tight, it is a condition of publication that you submit the revised version of your manuscript before 16-Mar-2019. Please note that the revision deadline will expire at 00.00am on this date. If you do not think you will be able to meet this date please let me know immediately.

on behalf of Dr Denise Greig (Associate Editor) and Kevin Padian (Subject Editor)
openscience@royalsociety.org

Associate Editor Comments to Author (Dr Denise Greig):

Associate Editor

Comments to the Author:

I would like to commend the authors for the trouble shooting they did to figure out and correct the manufacturer's mistake. I also very much appreciate the clarifications in the method section and the response to reviewers. Like Reviewer #2 notes, there are still a few leftover typos from the first version, but once those are corrected, this will be an interesting addition to the literature on marine animal locomotion.

Below are the typos I noticed:

Line 143. Delete the word "been"

Line 217. Correct "sing" to "using"

Line 218. Delete "was"

Line 243. Add "(CWT)" as you use this abbreviation in the next sentence? (if that is the correct interpretation)

Line 289-290. Change to "two CATS camera tags" (I think you used the CATS abbreviation above)

Line 380. Should this be Fish #1? As everything above was talking about Fish #2? Please be consistent and refer to either Bluefin #1 and #2 or Fish #1 and #2 and use whichever nomenclature you prefer in Figure 3 as well.

Table 3. Please add a space and an "s" to "363 descents" in the Table title.

Reviewer comments to Author:

Reviewer: 2

Comments to the Author(s)

I am glad that the authors rectified the speed data. Nonetheless, the paper needs a careful proof-reading, as 'old' (wrong) numbers have not been replaced throughout. I have marked a few places, but there can be more. Other than that, I recommend publication. It is a unique set of data.
Manuscript

1. L. 327-328: "swimming speed showed a non-linear relationship with tail-beat frequency for Fish #2 (Figure 3)."

2. L. 820. Caption to Figure 3. The speed (and stride length) were not updated

3. L. 903. Caption to Figure 6. The speed (2.5 m/s) was not updated

Supplementary

1. Figure S2 (Supplementary S2) should be replaced. It is the old one.

Data in Dryad

1. The abstract seems to reflect the 'old' speed calibration: to sustain 3 m/s, a tuna needs more than 1Hz.

2. The CATS descent data seems to reflect the old calibration. Please check.

3. Some of the columns need units and need explanation. For example, in CATS data: what are D_{min} and D_{max} ? What is the difference between TBF and TBF_Hz? (The latter is the reciprocal of the former, and hence the first TBF is probably the tail-beat period in seconds, and not the frequency). Is TBF in the second file the frequency or the period?

4. I would be happier to see representative time series (at least one full period) of yo-yo, slow speed burst-and-coast, high speed burst-and-coast, and steady swimming, rather than the averaged data appearing in Dryad (it can be down-sampled to 10 HZ with hardly any harm).

Having both would have been even better.

Miscellaneous comments

1. L. 282: "Velocity"

2. L. 294: "Cats"

3. L. 778 (Ref. [77]): I am not sure if this reference will pass the production people.

4. Figure 3: I would have added lines of constant stride length v/TBF for a reference.

5. I would have been more careful in interpretations of yo-yo and burst-and-coast strategies. Both can reduce the cost of transport only if (chemo-mechanical) propulsion efficiency is either thrust- or speed- dependent (Ref. [28], [65] and also the reference cited in footnote 2 of the first review). With constant propulsion efficiency, the swimming strategy that minimizes the cost of transport is constant-speed-constant-depth. In the context of these strategies, a few specific comments:

L. 406-410. This is a conjecture. I am not sure if the statement is supported by the data.

L. 435: During acceleration/ascent the fish 'pays' for the lack of activity during deceleration/descent by increasing its TBF, and therefore the frequency argument should probably be reworded. Adding the word 'red' before the word 'muscles' could solve most of the problem [65].

L. 473-476: The difference in energetic costs of ascent and descent is independent of speed. It is just the difference in the potential energy (submerged weight times the change in depth) divided by the propulsion efficiency.

L. 493: And also the flex at the caudal peduncle.

Author's Response to Decision Letter for (RSOS-190203.R0)

See Appendix C.

Decision letter (RSOS-190203.R1)

09-Apr-2019

Dear Dr Gleiss,

I am pleased to inform you that your manuscript entitled "Direct Measurement of Swimming and Diving Kinematics of Giant Atlantic Bluefin Tuna (*Thunnus thynnus*)" is now accepted for publication in Royal Society Open Science.

on behalf of Dr Denise Greig (Associate Editor) and Professor Kevin Padian (Subject Editor)
openscience@royalsociety.org

Follow Royal Society Publishing on Twitter: [@RSocPublishing](https://twitter.com/RSocPublishing)
Follow Royal Society Publishing on Facebook:
<https://www.facebook.com/RoyalSocietyPublishing.FanPage/>
Read Royal Society Publishing's blog: <https://blogs.royalsociety.org/publishing/>

Appendix A

The paper presents a unique set of data, but having great concerns about the accuracy (coherency) of this data, I am not in favor of publication. The following comments address mainly the data issue.

The data

1. The quoted speed values do not fit anything I know about fish locomotion. It is quite unlikely that fish # 1 (and, by extension, fish # 2) swam faster than 3 m/s at all times. It is equally unlikely that a fish (any fish) will use a burst-and-coast swimming strategy at high speed. Four different arguments to support the first claim follow immediately below. The second claim is recapitulated in item 3.

I will start with the basic data that will be needed to present the first three arguments. It is summarized in the following table:

parameter	source	value
fork length l (m)	table 1 of the paper	2.42
mass m (kg)	FishBase	228
effective diameter d (m)	from mass, assuming double-ogive shape	0.46
cross section area S (m ²)	$\pi d^2/4$	0.17

The mass was estimated from the fork length using a regression from FishBase ($a=0.0231$, $b=2.934$); the effective diameter and the cross section area were estimated from the mass estimate by assuming that the body has a shape of a double-ogive. Hydrodynamic characteristics (drag and power – parasite and induced) were estimated using the methods outlined in the supplementary material to Ref. [28]. Parasite drag estimate has accounted for five fins: a pair of pectorals, second dorsal, anal and caudal. Relevant dimensions were measured on a photograph found on the net.¹

Estimates of the induced drag (drag due to lift) were based on the assumption that most

1

https://www.google.co.il/search?hl=en&tbm=isch&source=hp&biw=1198&bih=714&ei=sFSIW4SoGYT6qwGpgqHIAw&q=bluefin+tuna&og=bluefin+tuna&gs_l=img.3..0110.2034.8639.0.8964.13.10.0.3.3.0.220.1281.0j4j3.7.0..0...1ac.1.64.img..3.10.1299....0. EV8z78lQil#imgrc=06xQrDy4w-t9UM:

of the lift is generated by the pectoral fins. The estimates are greatly affected by the excess density parameter (the ratio of the excess density to the density of water) k_{SF} , which is a priori unknown. The values in the following table have been based on $k_{SF} = 0.04$, the maximal value I deemed acceptable based on the lift coefficient (and the fact that large tunas have a swim bladder). Halving the excess density parameter would have reduced the induced drag four-fold, but it would have hardly affect the total drag, especially at high speed.

parameter	units	value		
		1.0	2.0	3.0
swim speed v	m/s	1.0	2.0	3.0
parasite drag coefficient C_{D0} based on cross section area S		0.12	0.115	0.1
induced drag coefficient K based on cross section area S		0.065	0.065	0.065
lift coefficient C_L based on cross section area S and $k_{SF} = 0.04$ (where k_{SF} is the relative excess density)		1.05	0.26	0.12
parasite drag at 15°C $D_0 = (1/2)\rho v^2 S C_{D0}$ (where ρ is the density of water)	N	13	36	75
total drag (parasite and induced) at 15°C with $k_{SF} = 0.04$ $D = (1/2)\rho v^2 S C_D$	N	16	37	76
hydrodynamic power associated with parasite drag $P = (1/2)\rho v^3 S C_D$	watt	16	75	227
initial deceleration with no tail beat $a = D/m(1+k_a)$ (where $k_a \approx 0.06$ is the added mass fraction)	m/s ²	0.068	0.15	0.31
sustained descent rate with no tail beat $v_d = P/mgk_{SF}$ based on $k_{SF} = 0.04$	m/s	0.18	0.84	2.54

When a fish stops powering its tail but continues to swim at the same depth, it must lose its kinetic energy. The deceleration can be estimated as the ratio of its drag $D = (1/2)\rho v^2 S C_D$ to its apparent mass $m(1+k_a)$. Estimating the added mass $m_a = k_a m$ at 6% of the real mass [27], the initial deceleration at 3 m/s should have been 0.32 m/s². Shutting the power off for 3 seconds (left panel in figure 4), should have reduced the swim speed by almost 1 m/s. I do not see a change of this magnitude on figure 4. It is possible that the speed measurement was simply too slow to capture speed variations on time scale of 1 sec, but then the speed should not have been displayed in the context of burst-and-coast swimming. Shutting the power off for 3

seconds at 1 m/s would have led to the loss of only 0.2 m/s in the swim speed.

When a negatively buoyant fish stops powering its tail but continues to swim at the same speed, it must lose its potential energy – in other words, it must go down. The required rate of descent can be estimated as the ratio of the hydrodynamic power needed to overcome drag $P = (1/2)\rho v^3 SC_D$ and the submerged weight $k_{SF}mg$. Perhaps counterintuitively, large k_{SF} make the descent rate needed to supply the power smaller. With $k_{SF} = 0.04$, the descent rate that would have been needed to sustain the swim speed of 3 m/s is 2.5 m/s. It can be interpreted as a 60° dive. It is not what we see. With $k_{SF} = 0.02$, the descent rate that would have been needed to sustain the 3 m/s exceeds the swim speed!

I can put it differently. A 1 min-35 m-dive shown on the right panel of figure 4 implies that the fish was descending at 0.6 m/s. With $k_{SF} = 0.04$, the rate of loss of the potential energy $k_{SF}mgv_d$ is 52 watt. This power can sustain a swim speed of 1.8 m/s (to sustain 3 m/s, k_{SF} should have been an impossible 0.16). Sinking at 0.6 m/s while swimming at 1.8 m/s can be interpreted as a 19° dive. With pitch attitude of 15°, this estimate is viable. I draw the authors' attention to the fact that the swimming direction will typically lie below the body axis.

Closely following [28] (actually, closely following reference [12] in [28]), and assuming body temperature of 20°C, I estimate the standard metabolic rate at 0.35 mmol ATP/s. Combined with hydrodynamic parameters listed in the preceding table (together with hydrodynamic propulsion efficiency of 0.7 and muscle efficiency of 24 Joule per mmol ATP) it yields approximately 1 m/s for the swim speed that minimizes the cost of transport. I could err two-fold in the standard metabolic rate, which would have yielded a 30% higher optimal speed, and I could err two-fold in the excess density parameter, which would have yielded a 20% lower speed - but I could not err three-fold in the optimal swim speed. It is unlikely that a fish (any fish) will swim 48 hours at two or three times the energetically-optimal speed.

Ref. [63] reports the stride length (the distance travelled over one tail-beat) for kawakawa (*Euthynnus affinis*) - a typical thunniform swimmer similar in appearance to the bluefin tuna. I recapitulated the relevant data from that reference in the following table. Stride length does not exceed 0.6 body length. I see this result as rather general, because basic biomechanical analysis (I will skip the one-page derivation here) shows that the optimal stride length for a thunniform swimmer should be, roughly, the same as the lateral travel of the caudal fin during a tailbeat (i.e. twice the lateral side-to-side amplitude of the caudal fin). Because the latter is limited by flexibility of the fish body, the stride length of 1.4 body lengths implied here (swimming 1.2 body length a second at 0.9 HZ) seems very unlikely. By the way, in view of the uncertainty in the speed calibration, I would have been very careful claiming a non-linear tail-beat-frequency-speed relation.

parameter	value	value	value	value	value
swim speed v (body lengths per second)	3.1	3.8	4	4.7	8.2
tail-beat frequency f (Hz)	7.7	9.1	10	12.5	14.3
tail amplitude A (side-to-side, body lengths)	0.16	0.26	0.25	0.2	0.34
stride length $\lambda = v/f$ (body lengths)	0.40	0.42	0.4	0.38	0.57

2. I do not understand why the authors did not calibrate the speed sensor ‘in vitro’ – if not before its deployment, at least after the recorder has been retrieved. Having printed the impeller implies that an identical copy can be printed again, and therefore the device can be replicated even if the original one has been damaged. Because the paper strongly relies on the speed data, I urge the authors to make the direct calibration and not to rely on an indirect one.

3. Burst and coast strategy is energetically justified only if chemo-mechanical propulsion efficiency depends on thrust or/and on the twitch frequency [68].² The propulsion efficiency can certainly change at low speeds because the caudal peduncle may not be flexible enough,

² “Locomotion of neutrally buoyant fish with flexible caudal fin,” Journal of Theoretical Biology 399, 2016, pp. 159-165, <http://dx.doi.org/10.1016/j.jtbi.2016.04.001>

but I can hardly see how it can change at 3 m/s, which is a large fraction of the maximal speed. The same remark applies to the locomotion efficiency of the yoyo diving.

Small comments (in no particular order)

1. It is possible that I missed it, but water temperature is nowhere to be found. It is an important parameter.
2. I think that the mass estimate should appear in the paper. It is important as well.
3. The quality of the figures is inadequate to extract any quantitative information. Either the quality of the figures should be improved, or the data should be made available as supplementary material.
4. I think that galvanic-timed release should have been tested in a lab before the claim is made on its malfunction. It is possible that a different lesson can be learnt from the delayed releases.

Appendix B

Subject Editor Comments:

We hope that the authors will be able to address the comments, recalibrate as necessary, and resubmit having addressed all reviewers' comments. Thanks for submitting.

Associate Editor Comments to Author (Dr Denise Greig):

These are fascinating data and it is clear that a lot of work went into this project. Deploying new biologging technology onto fast moving pelagic species is no small feat, however the methods and analyses need to be revised for clarity. Additionally one reviewer had serious concerns about the post-processing of the telemetry data. I hope you will be willing to address his in-depth concerns and resubmit this manuscript to RSOS.

We would like to thank the editor for providing us with the opportunity to revise the manuscript. Below we have given a point-by-point response to both referees. We have not provided a detailed rebuttal to the first part of the criticisms by referee #2, as related to the swimming speeds being impossibly high, as he is correct in every single point. We were suspicious of our speed estimates early on in this work ourselves and indeed, performed these calibrations and a number of validations (dive angles vs vertical velocities vs dive depths) to consistently receive the same results. Indeed, our good relationships of TBF and swimming performance also suggested our work was valid.

After careful reading of the criticisms of referee #2, however, we performed the analysis one more time just to get the same results. This suggested that the only remaining problem was in the hardware of the data-logger, since if the estimation of depth by the tag was incorrect, all our post-processing down-stream would be incorrect too. After consulting a detailed bathymetric map of the location where the fish was released, we found that the recorded depths was exactly twice as large as the seabed depth at that particular location. After approaching the manufacturer with this information, they conceded that at times 10 and 20 bar sensors were mixed up (apparently, their supplier provides these unlabelled). Thus it appears a 20bar sensor, rather than 10bar sensor was installed in our unit, which was then calibrated with the same parameters as a 10bar unit, thus reporting depth values that are twice as large. Because the error was introduced in the depth calibration, all speed calibrations appeared valid. We have recalibrated the sensor and adjusted all reported data in the manuscript. We hope the referee find the data now match the theoretical expectations and we look forward to his detailed comments about our manuscript.

We apologise to referee #1 that our speed estimates are now far more “ordinary”, than the previous estimates, although bluefin are still very fast given the temperatures.

My specific questions/comments below:

Great introduction!

Thank you.

Methods

The tag descriptions are confusing... I did not see anything about camera portion of the CATS tag when sat tag portion and casing were described (although it is pictured in Figure 1). What about the depth and temperature sensors – where on the tag are they located? Are they part of the Wildlife Computers and ATS portions of the tag? comparable between the two tag types? Programmed the same way?

We have moved the methods to follow the introduction. We agree that this has made it considerably easier to follow the narrative. We have also added more detail about the sensors and their sampling regimes. Lack of this information was an unnecessary oversight. Apologies.

Added to line 202:

CATS tags sampled tri-axial acceleration, depth and temperature (and additional other sensor data not used here) at 40 Hz. In addition to its suit of sensors, the CATS tags contained a digital camera facing the rear of the fish to validate signals from the accelerometers and to document the fishes' environment.

Added to line 222:

CEFAS G6as sensor array contained tri-axial accelerometers sampled at 30 Hz here, as well as temperature and depth sampled at 1 Hz.

Do you expect positively buoyant tag casings to affect your kinematic measurements?

This is a very good question. We did not perform quantitative tests of the buoyancy of the tags compared to those of tuna, but instead minimised excessive positive buoyancy through floatation tests. We have added the following to line 180:

Both types of tags were encased in positively buoyant foam to permit floatation following pop-off, potentially altering buoyancy of its carrier. During the design-process excessive positive buoyancy was minimised to reduce this effect. Completely assembled tags only showed marginal positive buoyancy as evidenced by only a very small fraction of the tags protruding above the water's surface while floating.

Line 441 please change parenthetical to (see Figure 1 for CATS tag and Whitmore et al for CEFAS G6a)

Done

Results – the results are confusing/missing some information. Table 2 is helpful for the methods and results, but in the text it is difficult to figure out which data were acquired from which fish. I think maybe having the Methods at the end adds to the confusion – is there a reason that you chose to do this?

We have moved the methods before the Results and feel this has helped with clarity.

Line 160 – first mention of the CATS tag (please add abbreviation) - Unclear how "glides" are defined (using the video)? Video mentioned in the Discussion but not Results Line 234 – this is fish #1 also correct?

Agreed that this was somewhat unclear as a result of the methods being placed at the end of manuscript. The section detailing the definition of glides from sensor data is now appears before the results section. This hopefully helps the reader to understand how passive periods were defined.

Line 453 and 459 – two undefined abbreviations (ABFT and GTR), GTR is defined later in Line 466 Line 459 – please add a references for the washers that are “previously described”

Done

Figure 1D – is the photo upside down?

The angle that the photo was taken in was awkward. If the picture is used right-way up and it appears even stranger than the upside-down version chosen. Apologies, but we have no other images to improve this.

Figure 3..I thought you only had data from Fish #1 (CATS) and Fish #2 (G6a)? This figure shows data from Fish #4?

This has been corrected

Reviewers' Comments to Author:

Reviewer: 1

Comments to the Author(s)

The authors deployed accelerometers and speed sensors on large Atlantic blue fin tunas to show their fine-scale swimming behavior in the wild. Although sample size is only two, I think the authors did a great job, given the difficulties in obtaining this kind of data from large, wild tunas (tag recovery points are very far). The data are new to science, and worthy of publication. I am especially impressed by fast cruising speed of tunas. My comments are minor, and I believe this paper will be a great contribution to Royal Society Open Science.

Thank you for your encouraging words. We hope you find the manuscript improved as a result of the review.

L43 Suggest deleting "a number of".

Done

L53 Delete comma after "Tunas".

Done

L77 "Up to".

Done

L159-165 Suggest moving this part to methods

The methods section has been moved from the end of the manuscript to following the introduction and this comment has been resolved as a result.

L168 CFL should be spelled out.

Done

L184-185 Suggest moving this sentence to methods

The methods section has been moved from the end of the manuscript to following the introduction and this comment has been resolved as a result.

L227 Suggest changing "dive angles" to "pitch angles" to avoid misunderstanding.

Done. Also replaced later on in the MS.

L294-295 Here I cannot catch the point. Why are fast and steep ascents related to increasing horizontal distance? What is "per unit of energetically intensive ascent"?

After reading this sentence again, we agree that some unpacking may be necessary. We have expanded the sentence in the following manner. We hope this somewhat clarifies the point.

Dives also feature marginally asymmetrical geometries, ascents are both faster and steeper than descents, thus *decreasing the time engaged in the energetically expensive portion of a dive and increasing the amount of time engaged in gliding phases at shallow angles, thus increasing horizontal distance traversed per unit energy.*

L326 "between ascents and descents"

Done

L391 Thermoregulation is functionally related to foraging in those fishes.

Indeed. We have clarified this point by adding the following to the sentence:

Certainly for Atlantic bluefin tuna and other species of tunas, thermoregulation has been a prominent explanation for vertical excursions, *permitting fish to feed below the thermocline while maintaining elevated muscle temperatures* [48, 49]

L406 Suggest deleting "a number of".

Done

L525-549 It's unclear to me why you didn't calibrate your paddle-whales in flow tanks (or something similar)? I am concerned that the pitch method doesn't work well for fishes due to shallow pitch angles in general (and that's why the relationships are so noisy?).

This is a very good question. My personal opinion, and I currently have no data to back this up, is that because of boundary layer issues, ingression of particles or non-standardised placement of tags on animals, *in situ* offer a better means for calibration. It is true that the shallow angles likely result in increased errors, but these are unlikely to bias the estimation, because error should be equal in over and underestimation (although variable angle of attack will cause some minor biases). The procedure we used results in a data-rich data-set, which I argue takes care of this problem. I think our very tight relationships between tail-beat frequency are evidence for the methods having worked well. Having said that, it would be a good methodological study in the future to establish which method is superior.

Referee #2

We would like to thank the referee for his very detailed comments. Indeed, his convincing argument is the reason we have approached the manufacturer (after redoing the analysis on four occasions and coming up with the same results). This led us to the insight that a mix-up in depth-sensor by the manufacturer has resulted in a mis-calibration, an overestimation of depth by a factor of two, in turn resulting in an overestimate in our calibration by the same factor. We have redone the analysis in light of the new depth calibration, adjusted all figures and graphics throughout the MS and hope this will take care of this major criticism. Specifically the referee also had a number of more detailed comments that were presents in his convincing argument.

Non-linear fit to TBF and speed

Apologies for this. Indeed, stride-length tends to be conserved across TBFs and thus the relationship between speed and TBF is generally found to be linear (as we have also found later in the MS, Fig. 5) and there was no particular reason we fitted this only that it looked marginally better. Linear fits were essentially just as good as the polynomial ($R^2=0.61$ vs 0.58). We have now only reported a linear fit. The reason for the minor kink in the low speed range of the graph could either be true signal: fish introduce more gliding during horizontal swimming at low speeds, similar to gliding being used as a means to control swimming speed during diving or that there are some minor problems in the calibration at the lowest speeds.

Small comments (in no particular order)

1. It is possible that I missed it, but water temperature is nowhere to be found. It is an important parameter.

It is indeed. We have now reported these in Figure 2 as a time-series.

2. I think that the mass estimate should appear in the paper. It is important as well.

Done. It has been added to Table 2

3. The quality of the figures is inadequate to extract any quantitative information. Either the quality of the figures should be improved, or the data should be made available as supplementary material.

All data contained in the figures of the paper will be contained in a Dryad repository, as was stated in the “data accessibility statement”. However, I am eager to hear how the referee suggests our figures could be improved for clarity to allow the extraction of quantitative information, bearing in mind that the correlations between the different variables were also reported in the MS?

4. I think that galvanic-timed release should have been tested in a lab before the claim is made on its malfunction. It is possible that a different lesson can be learnt from the delayed releases.

Apologies if we were not clear here in why we believe our pop-off mechanism failed. We do not think the GTR itself failed (these are off the shelf lobster pot releases and are as simple as they come, fail-safe one could say), but rather that the zip-tie that was passed through the fin got jammed in the grooves of the tag. The forces of the swimming fish were insufficient to bend the zip-tie and pull the tag off the fin. We do agree its probably a good idea to elaborate on this, since the new attachment is a large part of the MS. However, we are not 100% certain that this is the explanation for delayed pop-off. We have added the following sentence to line 311:

The reason for the delayed pop-off are not known, but we speculate that the stiff zip-ties were jammed in the grooves of the tag and the fin and only working itself loose following an extended period at liberty.

Appendix C

Associate Editor Comments to Author (Dr Denise Greig):

Associate Editor

Comments to the Author:

I would like to commend the authors for the trouble shooting they did to figure out and correct the manufacturer's mistake. I also very much appreciate the clarifications in the method section and the response to reviewers. Like Reviewer #2 notes, there are still a few leftover typos from the first version, but once those are corrected, this will be an interesting addition to the literature on marine animal locomotion.

Thank you.

Below are the typos I noticed:

Line 143. Delete the word "been"

Done

Line 217. Correct "sing" to "using"

Done

Line 218. Delete "was"

Done

Line 243. Add "(CWT)" as you use this abbreviation in the next sentence? (if that is the correct interpretation)

Done

Line 289-290. Change to "two CATS camera tags" (I think you used the CATS abbreviation above) Line 380. Should this be Fish #1? As everything above was talking about Fish #2? Please be consistent and refer to either Bluefin #1 and #2 or Fish #1 and #2 and use whichever nomenclature you prefer in Figure 3 as well.

Done. We have now used consistent terminology to refer to the individual animals and rectified all numerical issues.

Table 3. Please add a space and an "s" to "363 descents" in the Table title.

Done

Reviewer comments to Author:

Reviewer: 2

Comments to the Author(s)

I am glad that the authors rectified the speed data. Nonetheless, the paper needs a careful proof-reading, as 'old' (wrong) numbers have not been replaced throughout. I have marked a few places, but there can be more. Other than that, I recommend publication. It is a unique set of data.

Thank you for spotting those and thank you again for your thorough review.

Manuscript

L. 327-328: “swimming speed showed a non-linear relationship with tail-beat frequency for Fish #2 (Figure 3).”

Done and changed to linear.

L. 820. Caption to Figure 3. The speed (and stride length) were not updated

Now updated.

L. 903. Caption to Figure 6. The speed (2.5 m/s) was not updated.

Done

Supplementary 1. Figure S2 (Supplementary S2) should be replaced. It is the old one.

Done

Data in Dryad

The abstract seems to reflect the ‘old’ speed calibration: to sustain 3 m/s, a tuna needs more than 1Hz.

We were unable to change the abstract in Dryad because we are not creating a new submission. I am assured that once published the new abstract will be adopted.

The CATS descent data seems to reflect the old calibration. Please check. Some of the columns need units and need explanation. For example, in CATS data: what are Dmin and Dmax? What is the difference between TBF and TBF_Hz? (The latter is the reciprocal of the former, and hence the first TBF is probably the tail-beat period in seconds, and not the frequency). Is TBF in the second file the frequency or the period?

We have added a detailed description of all columns in an additional sheet and fixed the problem with the descent values. Again, well spotted. Thank you.

I would be happier to see representative time series (at least one full period) of yo-yo, slow speed burst-and-coast, high speed burst-and-coast, and steady swimming, rather than the averaged data appearing in Dryad (it can be down-sampled to 10 HZ with hardly any harm). Having both would have been even better.

We have added all data to Dryad that are necessary to validate our statistical analysis. Since we do not discuss the data in any detail beyond what is shown in Figure 4, we feel it is not necessary to share the raw data. Indeed, we have a student currently working on a more detailed account of the kinematics and would therefore rather not share data beyond what is published here.

Miscellaneous comments

L. 282: “Velocity”

Done

2. L. 294: “Cats”

We are not sure what the referee refers to here. We have written “CATS” in all caps, as it is an acronym.

L. 778 (Ref. [77]): I am not sure if this reference will pass the production people.
Well spotted. Thank you. We have fixed it.

Figure 3: I would have added lines of constant stride length v/TBF for a reference.
Done. Very good suggestion. Figure looks improved.

I would have been more careful in interpretations of yo-yo and burst-and-coast strategies. Both can reduce the cost of transport only if (chemo-mechanical) propulsion efficiency is either thrust- or speed- dependent (Ref. [28], [65] and also the reference cited in footnote 2 of the first review). With constant propulsion efficiency, the swimming strategy that minimizes the cost of transport is constant-speed-constant-depth. In the context of these strategies, a few specific comments:

We appreciate the referee's comments and acknowledge that perhaps we could have done a better job stating some of the assumptions of the hydrodynamic models we have used to present our argument. We have now added an additional paragraph to line 493 to state that these assumptions and that they have been tested in bluefin, but that caution needs to be exercised when transferring this to other species:

It is worth noting, that our discussions related to vertical movements providing an energy saving for fish rely on an important assumption in Weihs' theory [17], namely that the efficiency of transforming chemical energy into mechanical work must differ between ascent and descent [27]. Weihs' originally proposed that the drag incurred by a fish swimming with lateral undulations should be in the region of 2-3 times greater than those of a rigid, gliding fish. Therefore, if a fish can propel itself passively using gravity during descent, it would incur substantially less drag than if it were to swim continuously. Takagi et al. [55], using computational fluid dynamics quantified and confirmed the drag coefficient of juvenile Pacific bluefin tuna to be greater during undulatory swimming than gliding by a factor of two, further supporting that tuna may save energy through repeated diving. Some caution needs to be exercised in translating these data to other fishes, as this effect may not be universal and might depend on the kinematics of the species in question.

L. 406-410. This is a conjecture. I am not sure if the statement is supported by the data.
Agreed. This statement was poorly worded and as a result misrepresented the literature. We have changed it in the following manner:

This represents an important aspect of Weihs' two-stage locomotion model [17, 64], suggesting that some dives are performed to increase the economy of horizontal movements, which is in broad agreement with captive work on bluefin tuna [55, 56].

*This represents an important aspect of Weihs' two-stage locomotion model [17, 64], suggesting that some dives **could be** performed to increase the economy of horizontal movements, which is in broad agreement with captive work on bluefin tuna [55, 56].*

L. 435: During acceleration/ascent the fish 'pays' for the lack of activity during deceleration/ descent by increasing its TBF, and therefore the frequency argument

should probably be reworded. Adding the word ‘red’ before the word ‘muscles’ could solve most of the problem [65].

Thank you very good suggestion. We have added “aerobic” instead of red

L. 473-476: The difference in energetic costs of ascent and descent is independent of speed. It is just the difference in the potential energy (submerged weight times the change in depth) divided by the propulsion efficiency.

In principal that is true. However, if efficiencies are dependent on speed, then we should also see differences beyond simply the change in potential energy. Our data show clearly that the prevalence of gliding is strongly dependent on swimming speed, and since Takagi et al. have demonstrated that drag is greater by a factor of two during swimming, as opposed to gliding suggests that drag and by extension mechano-chemical efficiency during descents will depend on speed. This is confirmed in our data on tail-beat frequency and swimming speed.

We have adjusted this paragraph to reflect the basic tenet that in principal, only potential energy differs.

*Ln 461 At first hand, this may suggest that overcoming buoyant forces (i.e. **change in potential energy**) represents a minor component of the power required to swim.*

*Ln 471 Thus, at faster speeds, the relative cost of counteracting negative buoyancy becomes increasingly negligible compared to the costs related to parasite drag; subsequently at the highest speeds we would expect there to be little difference between the energetic costs of ascents and descents **beyond the difference in potential energy.***

L. 493: And also the flex at the caudal peduncle.

Agreed. We have changed the sentence in the following manner:

*Of course, fish can alter other kinematic parameters that accelerometers cannot easily allude too, namely tail-beat amplitude **or the stiffening of the caudal region**, which may be altered in response to changes in their hydrodynamic environment.*